# Treatment Options in Early Stage (Stage I and II) of Oropharyngeal Cancer: A Narrative Review

**DOI:** 10.3390/medicina58081050

**Published:** 2022-08-04

**Authors:** Giuseppe Meccariello, Andrea Catalano, Giovanni Cammaroto, Giannicola Iannella, Claudio Vicini, Sheng-Po Hao, Andrea De Vito

**Affiliations:** 1Otolaryngology and Head-Neck Surgery Unit, Department of Head-Neck Surgeries, Morgagni Pierantoni Hospital, Health Local Agency Romagna, 47121 Forlì, Italy; 2Otolaryngology Unit, University of Ferrara, 44121 Ferrara, Italy; 3Department of Otolaryngology Head and Neck Surgery, Shin Kong Wu Ho-Su Memorial Hospital, School of Medicine, Fu-Jen University, Taipei 111, Taiwan; 4Otolaryngology and Head-Neck Surgery Unit, Department of Head-Neck Surgeries, Santa Maria delle Croci Hospital, Health Local Agency of Romagna, 48121 Ravenna, Italy

**Keywords:** oropharyngeal carcinoma squamous cell carcinoma, early stages carcinoma, trans oral robotic surgery, human papilloma virus, intensity modulated radiation therapy

## Abstract

Objective: to show an overview on the treatments’ options for stage I and II oropharyngeal carcinomasquamous cell carcinoma (OPSCC). Background: The traditional primary treatment modality of OPSCC at early stages is intensity modulated radiation therapy (IMRT). Trans-oral robotic surgery (TORS) has offered as an alternative, less invasive surgical option. Patients with human papilloma virus (HPV)-positive OPSCC have distinct staging with better overall survival in comparison with HPV-negative OPSCC patients. Methods: a comprehensive review of the English language literature was performed using PubMed, EMBASE, the Cochrane Library, and CENTRAL electronic databases. Conclusions: Many trials started examining the role of TORS in de-escalating treatment to optimize functional consequences while maintaining oncologic outcome. The head–neck surgeon has to know the current role of TORS in HPV-positive and negative OPSCC and the ongoing trials that will influence its future implementation. The feasibility of this treatment, the outcomes ensured, and the side effects are key factors to consider for each patient. The variables reported in this narrative review are pieces of a bigger puzzle called tailored, evidence-based driven medicine. Future evidence will help in the construction of robust and adaptive algorithms in order to ensure the adequate treatment for the OPSCC at early stages.

## 1. Introduction

Oropharyngeal squamous cell carcinoma (OPSCC) at an early stage is generally associated with better oncologic outcomes regardless of treatment modality. Therefore, therapy should achieve long-term, disease-free survival while minimizing acute and late toxicities and complications. Furthermore, human papilloma virus (HPV)-positive OPSCC is characterized by unique biology and clinical behavior in comparison with HPV-negative OPSCC. Consequently, in the latest 8th edition of the American Joint Committee on Cancer (AJCC), a specific staging algorithm for HPV-positive OPSCC [1] has been identified, which better reflects its more favorable prognosis. Most importantly, it has markedly reduced the influence of ipsilateral lymph node metastases for the p16-positive disease, considering that staged patients, applying the 7th edition of AJCC as N0, N1, and N2a and N2b lymph node classifications, have been described to have similar survival rates. This resulted in a down-staging of a substantial number of patients previously classified as having stage III and stage IVa disease (T1-2N1-2b) in AJCC 7th ed. to stage I in AJCC 8th ed. Although this new definition of stage I HPV-positive OPSCC is a more accurate reflection of prognosis, it is important to note that these similar outcomes across patients with AJCC 8th ed. stage I disease were historically obtained using very different treatment options. Patients with AJCC 7th ed. stage I to stage II OPSCC generally have been treated with radiotherapy (RT) or surgery alone, whereas those with AJCC 7th ed. stage IVa disease (T1-2 N2a-N2b) typically receive multimodality therapy [2].

By contrast, HPV-negative patients are generally resistant to treatment modalities. Moreover, advanced OPSCC requires multimodal approaches, such as surgery with adjuvant RT or concurrent RT with salvage surgery. HPV-negative OPSCC patients are usually older, the lesions are larger, and overall survival is less likely when compared to HPV-positive OPSCC patients. Although there has been improvement in understanding the background of HPV-negative cases, resistance to chemo/radiotherapy is still a dead end, indicating the need for further studies. 

Patients with OPSCC have distinct risk factor profiles depending on the HPV status of their tumors. Sexual activity and marijuana exposure are independently associated with HPV-positive cancers, while tobacco smoking, alcohol abuse, and poor oral hygiene are associated with HPV-negative cancers. Moreover, potentially associated with HPV-positive OPSCC are factors related to socioeconomic status, such as higher income and higher education level. In addition, patients with HPV-positive and HPV-negative OPSCC have distinct survival outcomes, including a greatly increased 5-year survival for patients with HPV-positive tumors. OPSCC HPV-positive patients are usually non-smokers or light smokers, except for a subset of heavier smokers. This distinction is important because non- and light smokers have a better survival rate and lower risk of second primary malignancies than heavier smokers. Most OPSCC patients present with local or loco-regional disease; however, a low percentage of patients present with distant metastasis. However, HPV-negative tumors more frequently present with distant metastasis (M1 disease) [2]. HPV-positive OPSCC seemed to improved survival, even in the setting of distant metastatic disease, as compared with HPV-negative OPSCC [2].

Despite the different biology and staging of HPV-positive versus HPV-negative ones, treatment algorithms are still similar, with some exceptions. Resection of primary tumors with ipsilateral neck dissection (or bilateral if tumor approaches the midline) and definitive RT represent the two options against early stage tumors [3]. However, early stage OPSCC definition by NCCN guidelines on treatment matches stage I and II of AJCC staging for HPV-negative tumors. Instead, for HPV-positive OPSCC, early stage definition by NCCN refers to a subgroup of AJCC stage I and II: tumors lower than 4 centimeters in greatest dimension without lymph node involvement or with only single node metastasis smaller than 3 cm [1,2].

Hence, the decision-making process about operative versus non-operative treatment options is based on patient and tumor factors identification by means of medical history, physical examination, and imaging analysis.

## 2. Materials and Methods

### Literature Search Protocol

A comprehensive review of the English language literature was performed using PubMed, EMBASE, the Cochrane Library, and CENTRAL electronic databases. Three searches using the keywords (1) surgery OR robotic OR transoral, (2) chemotherapy OR radiation therapy OR chemoradiotherapy, and (3) oropharyngeal OR neck OR metastases were performed. These searches were combined with the AND function to find all relevant articles. The following inclusion criteria were applied to each article: (1) available information on outcome data; (2) data concerning the treatment options, HPV-status, TNM staging, preoperative radiologic work-up. Papers not meeting the inclusion criteria were excluded. Further exclusion criteria included case reports without significant data. To further reduce the risk of an incomplete literature search, a manual search through the references of the included papers was performed.

## 3. Discussion

### 3.1. Surgery

The surgical approach to OPSCC is neither easy for the surgeon nor easy to sustain for the patient. Open techniques bring along a set of complications and side effects that raise the morbidity of these procedures [4]. This type of surgery often includes a mandibulotomy, a tracheostomy, a pharyngotomy, and a flap reconstruction. Besides complications due to the sealing of the sutures (such as dehiscence and fistulae), there is a big impact on pharyngeal function in swallowing [5]. For these reasons, surgery of OPSCC has lost ground to less invasive treatment and was confined to the treatment of the advanced disease. The advancement of trans-oral techniques, such as Trans-oral Laser Microsurgery (TLM) and, especially, Trans-oral Robotic Surgery (TORS), determined the rivalidation of OPSCC surgical treatment [2]. The oncological and functional outcomes are preferable over open techniques [6]. However, TLM had disadvantages such as the poor visualization of the entire surgical field provided by the microscope and the low hemostatic efficacy of laser CO2 that required alternation with classical hemostasis techniques [7]. The TORS technique overcomes these limitations. Nowadays, surgery of OPSCC gains space in the treatment of the early stages alone or in combination with systemic therapy and/or RT [8]. Since 2009, after the approval of TORS by the federal drug administration (FDA), OPSCC surgeries entered into the “TORS era”, with a constant and positive trend for surgical therapy in the management of early stage OPSCC. There has been an increase in cases treated with surgery as a part of treatment and also in cases treated with surgery as a single treatment modality [5]. A prospective analysis of the oncologic and functional outcomes of TLM on 11 patients with tongue-based carcinoma stages I and II [9] showed only 1 regional failure 13 months after surgery in a patient that declined postoperative radiotherapy, while the other 10 had a complete R0 resection at 5 years. In this study, local control at 2 and 5 years were 100% for T1 tumors and 87% for T2 ones. A German retrospective study of 134 cases with T1-2 carcinomas of oropharynx that underwent TLM and unilateral or bilateral neck dissection [10] reported a disease-specific survival at 5 years of 78.6%, an overall survival (OS) of 59.9%, and a local control of 89%. A complete local resection R0 was obtained in 115 out of 134 cases (85,8%). Overall survival (OS) was significantly better for T1 in respect to T2 tumors and for R0 resection than R+, while disease-specific survival (DSS) was significantly better for N0 or R0 patients than N+ or R+ ones. Local control was reportedly higher for T0, N0, or R0 compared to T1, N+, or R+, respectively; however, the difference was not statistically significant [2].

A retrospective analysis of the National Cancer Database of United States [11] on long-term oncologic outcomes of TORS and TLM on T1 and T2 OPSCC (rate of HPV:19.2%) shows that TORS is linked to a lower rate of positive margins compared to non-robotic surgery and then a lower likelihood of adjuvant therapy. TORS is also associated with a lower hospital length of stay compared to TLM and non-robotic surgery. In a prospective trial [12] on stage I and II HPV-negative OPSCC, TORS associated with unilateral or bilateral neck dissection is a feasible single therapy to de-intensify treatment for HPV-negative OPSCC. DSS and OS at 29 months was 89.6% and 93.8%. Amongst the several open approaches, the most conservative appears to be lateral pharyngotomy, sparing the mandibulotomy. However lateral pharyngotomy for early OPSCC showed a disease-free survival at 5 years of 86% [13], lower than TORS, and a higher rate of complications (38%), mainly pharyngocutaneous fistula, hemorrhage or hematoma, infections, and pneumonia related to a longer hospital stay and longer persistence of tracheostomy and nasogastric feeding tube.

#### 3.1.1. Preoperative Patient Selection

The choice of the right treatment for each patient, in the optic of a tailored management, is fundamental to obtain the best result. Many patient factors could influence the final decision for surgical selection, such as a patient’s comorbidities, which should be analyzed by means of a detailed history. Surgery is preferred for active smokers unwilling to reform because of the reported limited radiotherapy efficacy and the effect of post-radiotherapy increasing risk for cerebrovascular diseases and second malignancies [14,15,16,17]. Furthermore, it should take into account the radiotherapy feasibility for the patient during a protracted course of RT, such as the distance from the treating center. In these circumstances, these patients may be better served by the surgical option [18,19]. Previous head and neck RT (most commonly lymphoma or skin cancer) leads to a choice surgery above new radiotherapy. Finally, patients with poor social conditions could not be compliant to radiotherapy schedules in comparison with surgical ones.

By contrast, definitive RT represents the first choice for patients with early-stage tonsil cancer with a medial retropharyngeal internal carotid artery (ICA) position, which increases the risk of postoperative bleeding [20,21]. The same is true for patients under anticoagulation therapy for high-risk medical events [22]. Even if uncommon in the early-stage patient, the presence of trismus is a contraindication for the transoral approaches. In the cases of definitive RT, or if the HPV-negative patient refuses RT, an open approach could be a valid treatment option [13].

TORS represents the first treatment choice in the presence of an oropharyngeal T1-2 exophytic primary tumor with minimal invasion and with high probability to achieve negative margins. The detection of cervical limfoadenopathy at presentation may suggest a neck dissection in order to exclude a nodes metastasis. The literature data reported pathologic downstaging in the N1 neck in approximately 30% of patients, whereas pathologic upstaging may occur in 30–40% of N0 patients [23,24].

The choice of the trans-oral resection of early-stage disease (T1-2N0) versus primary RT as a single-modality therapy represents a point of debate [25]. Rough measures of functional outcomes such as feeding tube dependence appear similar between TORS alone (0–7%) [26] and modern IMRT (4%) [27]. However, performing TORS allows a dose reduction in the radiotherapy, accompanied by a lower rate of acute and late side effects, compared to full-dose-definitive RT. The ORATOR trial [28] showed that oncological outcomes between RT and TORS plus neck dissection (followed or not by chemoradiotherapy) are similar in terms of OS and progression free survival, while the toxicity profiles are different. Anorexia, dry mouth, dysphagia, oral mucositis, nausea, odynophagia, taste alteration, and weight loss are adverse events shared between the two groups. Vomiting, hearing loss, tinnitus, sore throat, dermatitis or rash, neutropenia, sore mouth, fatigue, dysgeusia, constipation, and alopecia were more frequent in the radiotherapy group, while cough, other pain, weakness, and trismus were more common in the TORS group. Nonetheless, swallowing-related quality of life was improved in the RT group compared with the surgery group, even though it was not clinically meaningful.

Another tumor factor is the involvement of larger tumors of muscular or soft palate, which may encourage a non-operative approach. An early-stage patient rarely requires a reconstruction with local or free flaps; however, in the optic of better Quality-of-Life (QoL), RT and avoidance of reconstruction may be the best choice for them. Considering concerns about muscular invasion, closed surgical margins, and the inferior outcomes obtained with TORS, cases of uncommon primaries of the soft palate, posterior oropharyngeal, or the hypopharyngeal wall are areas may be better dominated by RT [14]. HPV relation is important since the early stage of OPSCC HPV-positive tumors recur locoregionally (after either surgery or radiotherapy) very infrequently, and best efforts for single-modality treatment should be take into consideration; however, some authors have demonstrated no prognostic significance for HPV+ in the overall or in the disease-free survival analysis for non-tonsillar, non-tongue-based SCC [15].

#### 3.1.2. Neck Dissection

Neck spreading of malignant cells from OPSCC to lymph nodes is an early and common event. Seventy-six percent of cases of tonsil cancer, more than 70% of tongue-based cancers, and from 25 to 74% of posterior pharyngeal wall cancers present with neck involvement [2,29]. The extension of the neoplastic disease to the regional lymph nodes is the most important independent factor for prognosis and is also related to survival [30]. Thus, whether to approach the neck and how much the dissection should be extended became an open question. In OPSCC, while the literature is not clear [31], it seems that neck dissection is linked to the overall survival of patients with the early stages of OPSCC [11]. Neck dissections were initially performed in a wide manner, such as the Radical Neck Dissection (RND), which includes the dissection from level I to level V with the sacrifice of the sternocleidomastoid muscle, internal jugular vein, submandibular gland, and spinal accessory nerve. The sacrifice of the latter was the main reason for morbidity in the patients undergoing RND, as it brings shoulder pain and functional impairment with shoulder drop. For this reason, nerve-sparing surgery was born [32], and led the way to the selective neck dissection (SND). Nowadays, SND of levels II–IV [33] is the standard method of managing neck treatment in OPSCC, leaving the use of RND only for advanced neck disease. Level II is the most frequent site of regional disease and, of 88 patients with OPSCC treated with TORS and SND at the University of Washington [34], only one patient had regional recurrence in level II, and no recurrence at all was found at levels I and V.

Some have questioned the need for neck dissection in clinically negative necks. However, the rate of occult metastases cannot be neglected (23 to 43%), and imaging is not sensible enough to identify them [35]. Ipsilateral neck dissection without clinically evident neck disease gives an improved outcome [36]. Attending a contralateral neck dissection in a clinically negative neck does not seem to be correlated to an increase in OS or relapse-free survival [37].

The nodal yield during neck dissection for early stages of OPSCC is not clearly associated with survival, especially in patients with two or more positive nodes [38]. However, in patients with very limited neck involvement (from zero to one positive node), harvesting at least 26 nodes may give an advantage on overall survival. Lymph node ratio (LNR), the ratio between positive nodes and total nodes examined, appears to be a prognostic factor in HPV-positive OPSCC. Specifically, an LNR equal or lower than 10% is linked to better OS and DSS at 5 years [39].

### 3.2. Radiotherapy

RT can be administered as adjuvant after surgery or as definitive treatment alone.

In the early stages, definitive RT is a grade 2A intervention for the NCCN guideline [40], meaning that there is a uniform consensus on this statement. Only in the HPV-positive OPSCC with a single neck metastasis larger than 3 cm or with 2 or more ipsilateral neck nodes is it recommended; it is not recommended alone but with concurrent systemic therapy [2].

The NCCN guideline [40] gives a grade 2B to the concurrent systemic therapy plus RT for the treatment of patients with early stage OPSCC with initial neck involvement (cN1), meaning that there is consensus on the appropriateness of the intervention, though it is not uniform. This statement regards both HPV-positive and HPV-negative OPSCC; however, for the first one, if the single neck metastasis is larger than 3 cm or there is 2 or more ipsilateral neck nodes, the NCCN guideline gives a grade 2A to the concurrent systemic therapy plus RT [40].

A review of the literature gives this intervention a category 2A, regardless of HPV status, because of the lack of high-quality prospective clinical evidence [40]. Adjuvant radiotherapy alone or in combination with chemotherapy is indicated when the pathologist finds high risk markers, such as extracapsular spread and a positive margin and no feasible revision [40,41]. A retrospective study of the National Cancer Database of the US, from 1998 to 2011, without stratification for HPV status, about patients with early-stage palatine tonsil SCC, found that multimodal treatment ensures the greatest survival at 5 years [42]. Surgery followed by adjuvant RT was better in 5-year OS (81.1%) than surgery alone and RT alone (67.4% and 63.4%, respectively), while no statistical difference in survival was found between RT alone and surgery alone. The worst survival outcomes were obtained when surgery alone did not manage the neck. Leaving out these inadequate surgeries from the surgery alone group, the OS was higher than the RT alone group. Another retrospective analysis of the same database has been published, albeit from 2010 to 2013 and selecting only stage I HPV-positive OPSCC with low to intermediate risk (excluding positive margins or macroscopic extranodal extension) [43]. They found that adjuvant RT, after surgery of the primary site and adequate neck dissection (nodal yield at least 15), does not carry a benefit in OS at 4 years for both low-risk and intermediate-risk patients.

Adjuvant RT for unilateral disease can be administered only ipsilaterally, even in patients with N2b lymph node stage, since OS, progression-free survival, and locoregional control are all reportedly higher than 90% [44]. Instead, sparing the primary site after adequate TORS resection and irradiating only the neck does not give any advantage in toxicity and clinical outcome [45].

Since HPV-positive patients have a better prognosis, QoL and toxicity of treatments are a main concern [46]. Definitive RT has shown better QoL outcomes than TORS with neck dissection in the randomized phase II ORATOR trial [28]. RT carries along less morbidity than surgery, but it is not free of side effects. Common side effects, affecting QoL, are represented by long-lasting mucositis and sore mouth [47], xerostomia, which can last more than 12 months [48], dysgeusia, dysosmia [49]. Dysphagia, dental disease, osteoradionecrosis, myelopathy, and trismus [50] are worse complications linked to RT. Notably, the majority of these side effects are dose-related. It is well known that the cumulative effect of radiation on pharyngeal constrictor muscles leads to long-term swallowing impairment; this risk is of 50% when the 78% of the cricopharyngeus muscle receives more than 60 Gy [51]. Tracheostomy and gastrostomy are sometimes required after TORS and after IMRT. The rate of tracheostomy dependency at one year varies among the studies; however, it is lower for TORS than for IMRT. The same is said for the feeding tube dependency, the persistence of which is linked to older age, higher N stage, pack-per-year smoking history, or concurrent chemotherapy [52].

All the above reasons, accompanied by the knowledge that HPV-positive disease shows a good response to therapy, led to the development of deintensification therapies to reduce toxicity while ensuring the same clinical outcome [2]. IMRT plus systemic therapy is the result of this process, improving the clinical sustainability of RT [6]. IMRT allows for the focusing of the therapy on the planning target volume (primary site plus a safety volume to account for microscopic extension of the disease and setup variations) and lymph nodal regions II–IV sparing the submandibular gland and swallowing structures, reducing the morbidity burden [53]. Studies reported a reduction in xerostomia from 36% to 3% and a reduction in PEG dependence at 6 months from 30% to 3% [54,55]. When extranodal extension is present, the use of RT alone in order to reduce morbidity is dangerous because it may be associated with worse outcomes in the long run [56]. 

Finally, proton-beam RT allows delivering the radiation dose to the target volume, stopping it behind this one and minimizing radiation to healthy tissues. This technique is still nascent, not widely available, and there is not yet a consensus on its cost-effectiveness ratio; however, it appears to reduce the total morbidity of the radiation therapy [53].

### 3.3. Chemotherapy

In stage I and II OPSCC, chemotherapy can be administered as adjuvant or neoadjuvant therapy (after or before surgery), as concurrent to RT, as induction therapy before RT, or alternating with RT [8]. However, the protocol must be tailored for the patient, particularly regarding their performance status.

Cisplatin is the cardinal of the systemic therapy and, as adjuvant therapy after surgery, it is the only recommended drug [2]. Carboplatin or cetuximab can be used instead of cisplatin when concurrent to RT for recurrent or persistent disease. Texans or 5-FU can also be used in induction or sequential systemic therapy. After induction, a radiotherapy approach can be used for high-dose therapy along with higher collateral effects. Carboplatin or 5-FU appears a feasible choice along with cisplatin in the primary systemic therapy plus radiotherapy [2].

Concurrent chemotherapy (administered within 7 days at the start of radiotherapy) improves the survival of patients with stage I OPSCC HPV-positive with positive lymph nodes compared to radiotherapy alone [57]. Otherwise, in stage I without lymph node extension, there is not an advantage in survival rate.

The most frequent side effects related to the chemotherapeutic drug are nausea and vomiting [47]. Dysgeusia, dysosmia, and xerostomia are common side effects linked not only to RT, but also to systemic chemotherapy [49].

The balance between what the patients need and what the patient can sustain is the key factor in the choice of chemoradiotherapy.

### 3.4. Adjuvant Therapy in the Post-TORS Early-Stage OPSCC

TORS resection should achieve surgical margins of tumor-free and reduced functional consequences. It is not uncommon to observe free surgical margins of 1–2 mm even after a complete transoral resection due to the permanent fixation process, which could reduce a histological specimen by approximately 30% [58]. Furthermore, the precise definition of a “close” margin remains unclear, even though it represents one of the main indication for post-operative RT [2], ranging from 1 to 5 mm [28,59]. For instance, the University of Pennsylvania trial is analyzing the possibility to consider a free primary site if the surgical margins are clear by 2 mm or more and if the lymphovascular space invasion (LVSI) or microscopic perineural invasion (PNI) are not reported [60]. LVSI and PNI represent risk-factors for locoregional recurrence; however, their impact on local versus regional recurrence is not clear, and neither is the RT indication [61].

The single node involved in the N1 neck stage after neck dissection is another controversial point of debate, considering that many OPSCC patients will obtain cure after an appropriate neck dissection, avoiding the addition of postoperative RT [62,63]. However, clinical N1 patients were also enrolled in several non-operative trials for locally-advanced stage III [64,65,66] and could benefit from postoperative RT, as reported in the inclusion criteria of the ORATOR and NRG/RTOG 1221 TORS trials [22,28]. Therefore, the choice of surgical treatment in this setting should be made after a multidisciplinary patient’s assessment, in which postoperative RT will be performed if extracapsular extension or additional nodal disease is detected. The ExtraNodal Extension (ENE) represents a negative prognostic factor, and these patients could obtain an improvement in survival rate from additional chemotherapy [67]. The degree of ENE, which needs postoperative combined RT and chemotherapy is not clear, even though the presence of microscopic ENE extending up to 1 mm or less from the nodal capsule could be well-cured by means of radiotherapy alone [68]. Therefore, OPSCC patients treated with ENE-positive TORS resection should be treated with postoperative RT. 

Furthermore, the ideal RT dose for HPV-positive tumors is still an open question, both for definitive and post-TORS radiotherapy. The ECOG 3311 and PATHOS trials [59,69] are currently studying the safety and efficacy of de-escalation from 60 Gy to 50 Gy in the post-TORS patients. Similarly, the NRG HN002 trial [70] is analyzing the dose-reduced definitive RT with or without chemotherapy in a group of HPV-positive OPSCC patients. If these trials demonstrate the safety and effectiveness of these protocols, the morbidity of both approaches could significantly decrease.

## 4. Conclusions and Future Perspectives

The conventional first-line treatment of OPSCC at early stage is IMRT. TORS could represent a valid alternative. Patients with HPV-positive OPSCC have superior overall survival after primary IMRT and report longer functional sequelae related to their disease treatment. Therefore, many trials have begun examining the role of TORS in de-escalating treatment to optimize functional outcomes, maintaining oncologic outcome [71].

New trials are ongoing, such as the ORATOR II trial, which aims to decrease the intensity of RT and chemotherapy to improve QoL, maintaining survival rates and avoiding triple-modality treatment. Finally, the PATHOS trial [59] is recruiting patients with early stages of HPV-positive OPSCC to compare standard versus de-escalated therapy. 

Furthermore, a significant breakthrough has been achieved in cancer immunotherapy. Among the Cancer-Immunity Cycle steps, the PD-1/PD-L1 checkpoint axis is most widely studied, which prevents the over-activation of T cells from damaging normal tissues and leads to the potential of tumor immune escape. The phase Ib trial published in 2016, KEYNOTE-012 (NCT01848834), was the first study investigating PD-1 blockade therapy in 104 recurrent/metastasis HNSCC patients expressing PD-L1 (38% were HPV-positive and 62% were HPV-negative). The overall response rate (ORR) reached 18% (95% CI, 8–32%), and median OS was 13 months [15]. Nevertheless, the notable success of PD-1/PD-L1-targeted therapy has been studied. Lenvatinib is a tyrosine kinase inhibitor of several VEGF receptors and could modulate immune suppression in the tumor micro-environment by angiogenesis inhibition. The effectiveness of pembrolizumab in combination with lenvatinib in patients with HNSCC has been supported by a phase II trial (NCT02501096) towards 137 patients with various advanced solid tumors (22 patients suffered HNSCC). The ORR at week 24 at the recommended dose (lenvatinib 20 mg/d, pembrolizumab 200 mg every 3 weeks) of HNSCC patients was 36% (8/22; 95% CI, 17.2% to 59.3%) [72]. Further, immunotherapy in HPV-positive OPSCC is also studied. Given the essential roles E6 and E7 play in HPV-positive cancers, they are usually selected as targets for a therapeutic vaccine. A phase I study (ACTRN12618000140257) assessed the safety, tolerability, and immunogenicity of an HPV E6/E7 vaccine (AMV002) in patients with HPV-positive OPSCC [73]. The vaccine-induced response rate was 83.3% (10 of 12). In addition, a phase Ib/II trial (NCT03162224) evaluating the safety and efficacy of MEDI0457 (DNA vaccine targeting HPV-16/18 E6/E7 antigens accompanied with an IL-12 adjuvant) plus durvalumab in HPV-positive R/M HNSCC patients is underway [74]. Currently, there are few studies on immunotherapy towards HPV-negative OPSCC. DURTRERAD is a randomized phase II trial evaluating feasibility and efficacy of durvalumab (D arm) versus durvalumab and tremelimumab (DT arm) in combination with radiotherapy as primary treatment for locally advanced HPV-negative HNSCC, with more than half being OPSCC [75]. Thus, although the monotherapy of novel agents has proved effective, while combinations of immunotherapy with conventional therapies and dual immunotherapy are undergoing clinical investigation, those management are, nowadays, reserved for recurrent/metastastic OPSCC. 

Finally, the head–neck surgeon has to know the current role of TORS in HPV-positive and -negative OPSCC and the ongoing trials that will influence its future implementation. The feasibility of this treatment, the outcomes ensured, and the side effects are key factors to consider for each patient. The variables reported in this chapter are pieces of a bigger puzzle called tailored, evidence-based-driven medicine. Future evidence will help the construction of robust and adaptive algorithms in order to ensure adequate treatment for the early stages of OPSCC.

## Data Availability

Not applicable.

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
