# Peer review of "Treatment Options in Early Stage (Stage I and II) of Oropharyngeal Cancer: A Narrative Review"

_medicina, 2022, doi:10.3390/medicina58081050_

Round 1

Reviewer 1 Report

Thank you for letting me review this interesting review.

1. It is important - when referring to outcome of different studies - to report how many patients or percentage of the series that consisted of HPV+ and HPV- tumors, respectively.

2. In the different studies it is important to declare outcome for patients who have been treated with surgery only separately from those who have had combined treatment since the point of reintroducing surgery for these patients is to spare them (the side effects of) radiotherapy and if a major percentage of the patients needed RT anyhow to achieve the results presented the surgical intervention may be questioned.

3. The english language needs to be revised, here are a few examples that need correction:

r 83: NPV-neg should be HPV-neg

r 126: patients should be a patient

r 168: ftransoral should be transoral

r 170: options should be option

r 173: linfadenopati should be lymfadenopati

r 201: cosniderations should be considerations

4. r 75: "HPV neg tu more frequently present with distant met" - needs a ref

5. r 197: "in case of uncommon primaries of the soft palate, posterior oropharyngeal or 197 hypopharyngeal wall are areas which may be better dominated by RT" please read and consider the following ref which shows that non tonsillar/non base of tongue cancer oropharyngeal cancer is different and that HPV presence here has limited value:

Prevalence of human papillomavirus and survival in oropharyngeal cancer other than tonsil or base of tongue cancer.

Marklund L, Näsman A, Ramqvist T, et al Cancer Med. 2012 Aug;1(1):82-8.

Author Response

All the Authors thank the reviewer for the comments and suggestions.

Question 1: It is important - when referring to outcome of different studies - to report how many patients or percentage of the series that consisted of HPV+ and HPV- tumors, respectively.

Reply:

We made corrections when possible due to difficulties in extrapolating data from each study

Question 2: In the different studies it is important to declare outcome for patients who have been treated with surgery only separately from those who have had combined treatment since the point of reintroducing surgery for these patients is to spare them (the side effects of) radiotherapy and if a major percentage of the patients needed RT anyhow to achieve the results presented the surgical intervention may be questioned.

Reply:

This issue is truly important, however, it has been too hard to extrapolate from some studies the correct proportion of different approaches and combinations. When possible we added that information requested. Unfortunately, it is a consistent bias in several studies, especially among the older.

Question 3: The english language needs to be revised, here are a few examples that need correction:

r 83: NPV-neg should be HPV-neg

r 126: patients should be a patient

r 168: ftransoral should be transoral

r 170: options should be option

r 173: linfadenopati should be lymfadenopati

r 201: cosniderations should be considerations

r 75: "HPV neg tu more frequently present with distant met" - needs a ref

r 197: "in case of uncommon primaries of the soft palate, posterior oropharyngeal or 197 hypopharyngeal wall are areas which may be better dominated by RT" please read and consider the following ref which shows that non tonsillar/non base of tongue cancer oropharyngeal cancer is different and that HPV presence here has limited value:

Prevalence of human papillomavirus and survival in oropharyngeal cancer other than tonsil or base of tongue cancer.

Marklund L, Näsman A, Ramqvist T, et al Cancer Med. 2012 Aug;1(1):82-8.

Reply:

We made corrections and implemented the text as requested

Reviewer 2 Report

The study tried to show an overview on the treatments’ options for the stage I and II OPSCC.

The content of this study is in-depth, but there are still some deficiencies.

Major issues:

In addition to surgery, chemotherapy and radiotherapy, the authors also need to analyze the potential of immunotherapy and targeted therapy in patients with OPSCC.

Author Response

All the Authors thank the reviewer for the comments and suggestions.

The study tried to show an overview on the treatments’ options for the stage I and II OPSCC.

The content of this study is in-depth, but there are still some deficiencies.

Major issues:

In addition to surgery, chemotherapy and radiotherapy, the authors also need to analyze the potential of immunotherapy and targeted therapy in patients with OPSCC.

Reply:

We added some information about novel immunotherapy and targeted therapy albeit those therapies are still in the evaluation and are reserved for advanced/recurrent cancers. Not being the chapter’s topic, we preferred just to give a little overview on this issue.